# MocoSFL: enabling cross-client collaborative self-supervised learning

**Jingtao Li** [*]
Arizona State University
jingtao1@asu.edu

**Lingjuan Lyu**
Sony AI
lingjuan.lv@sony.com

**Daisuke Iso**
Sony AI
daisuke.iso@sony.com

**Chaitali Chakrabarti**
Arizona State University
chaitali@asu.edu

**Michael Spranger**
Sony AI
michael.spranger@sony.com

## Abstract

Existing collaborative self-supervised learning (SSL) schemes are not suitable for cross-client applications because of their expensive computation and large local data requirements. To address these issues, we propose MocoSFL, a collaborative SSL framework based on Split Federated Learning (SFL) and Momentum Contrast (MoCo). In MocoSFL, the large backbone model is split into a small client-side model and a large server-side model, and only the small client-side model is processed locally on the client's local devices. MocoSFL has three key components: (i) vector concatenation which enables the use of small batch size and reduces computation and memory requirements by orders of magnitude; (ii) feature sharing that helps achieve high accuracy regardless of the quality and volume of local data; (iii) frequent synchronization that helps achieve better non-IID performance because of smaller local model divergence. For a 1,000-client case with non-IID data (each client only has data from 2 random classes of CIFAR-10), MocoSFL can achieve over 84% accuracy with ResNet-18 model. Next we present TAResSFL module that significantly improves the resistance to privacy threats and communication overhead with small sacrifice in accuracy for a MocoSFL system. On a Raspberry Pi 4B device, the MocoSFL-based scheme requires less than 1MB of memory and less than 40MB of communication, and consumes less than 5W power. The code is available at https://github.com/SonyAI/MocoSFL.

## 1 Introduction

Collaborative learning schemes have become increasingly popular, as clients can train their own local models without sharing their private local data. Current collaborative learning applications mostly focus on supervised learning applications where labels are available (Hard et al., 2018; Roth et al., 2020). However, availability of fully-labeled data may not be practical since labeling requires expertise and can be difficult to execute, especially for the average client.

Federated learning (FL) (McMahan et al., 2017) is the most popular collaborative learning framework. One representative algorithm is **"FedAvg"**, where clients send their local copies of the model to the server and the server performs a weighted average operation (weight depends on the amount of data) to get a new global model. FL has achieved great success in supervised learning, and has been used successfully in a wide range of applications, such as next word prediction McMahan et al. (2017), visual object detection for safety Liu et al. (2020), recommendation Wu et al. (2022a;b), graph-based analysis Chen et al. (2022); Wu et al. (2022c), etc.

For collaborative learning on unlabeled data, prior works (Zhang et al., 2020; Zhuang et al., 2021; 2022) combine FL scheme with classic self-supervised learning (SSL) methods such as BYOL (Grill et al., 2020) and Moco (He et al., 2020). These methods can all achieve good performance when clients' data is Independent and Identically Distributed (IID) but suffer from poor performance

---

[*]Work done during internship at Sony AI. Corresponding to: Lingjuan Lyu.

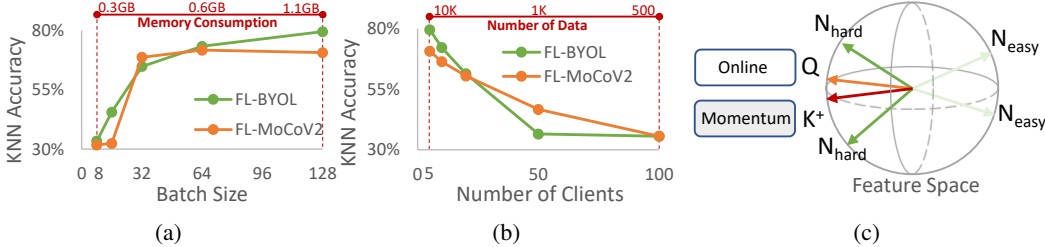

Figure 1: Challenges in FL-SSL schemes. (a) A large batch size is necessary to achieve good performance (KNN validation accuracy (Wu et al., 2018)) (b) Accuracy reduces with an increasing number of clients since the amount of local data is now smaller. (c) Hard negative keys are essential for the success of contrastive learning.

in non-IID cases. Recently, Zhuang et al. (2022) mitigated the non-IID performance drop with divergence-aware aggregation technique and provided state-of-the-art (SoTA) accuracy performance using a combination of FL and different SSL methods.

However, these SoTA FL-SSL schemes are not practical for cross-client applications. **First**, FL-SSL imposes a significant computation overhead and large memory requirement on clients. This is because SSL requires a large backbone architecture (Chen et al., 2020a) together with a large batch size to ensure good performance. As shown in Fig. 1(a), accuracy drops dramatically when batch size is low for both FL-SSL methods. **Second**, FL-SSL schemes fail to maintain a good accuracy when the number of clients is large (cross-client cases), as shown in Fig. 1(b). For a dataset with fixed size, when the number of clients increases, data per client decreases, resulting in accuracy degradation. The drop in accuracy is mainly because of the failure to meet data size requirement in performing contrastive learning. Zhang et al. (2020); Wu et al. (2021) attempt to address this issue in FL-SSL with remote feature sharing. However, this introduces a high communication overhead due to client-to-client feature memory synchronization; for a 100-client system, it costs around 2.46GB per synchronization per client.

To solve these challenges, we propose MocoSFL, a scheme based on Split Federated Learning (SFL) Thapa et al. (2020) that incorporates the feature memory bank and momentum model designs of MoCo (He et al., 2020). We adopt the SFL scheme for three reasons: (i) SFL utilizes a smaller client-side model and so reduces the computation overhead and has lower memory consumption and model parameters; (ii) SFL's latent vector concatenation enables a large equivalent batch size for the centralized server-side model, making micro-batch training possible for clients and thus reducing client's local memory; (iii) When combined with MoCo's key-storing mechanism, SFL's shared server-side model enables effective feature sharing, which removes the requirement of large amount of local data and makes the scheme possible for cross-client applications. As a result, MocoSFL achieves good accuracy with ultra-low memory requirements and computation overhead, and can support a very large number of clients. MocoSFL shows better non-IID performance since local model divergence is smaller. However, the use of SFL brings extra communication overhead as well as data privacy concerns. Thus, we present target-aware ResSFL (TAResSFL) module as an effective solution to mitigate these issues with small accuracy drop.

Our main contributions are:

- We identify two major challenges in deriving high accuracy in FL-SSL schemes for cross-client applications. These are availability of a large amount of data that is required for contrastive learning and the ability to process them in clients who may not have sufficient hardware resources.

- We propose MocoSFL, an SFL-based scheme to address the two challenges. MocoSFL uses a small client-side model, latent vector concatenation, and feature sharing. For cross-client case, MocoSFL is the only viable and practical solution. For cross-silo case, MocoSFL can achieve even better performance than SoTA FL-SSL schemes under non-IID setting because of smaller model divergence.

- To address communication overhead and privacy issues that are inherent to SFL-based schemes, we propose target-domain-aware ResSFL (TAResSFL) that effectively reduces the communication cost and mitigates model inversion attacks.

## 2 BACKGROUND

### 2.1 SELF-SUPERVISED LEARNING

To learn from unlabeled data, SSL schemes based on contrastive learning such as SimCLR Chen et al. (2020a), BYOL Grill et al. (2020), Simsiam Chen and He (2021) and MoCo He et al. (2020) have achieved great performance on popular benchmarks. Unlike other schemes (BYOL, SimCLR, etc.) that use other samples in the current data-batch as negative keys, MoCo uses previously computed positive keys as negative keys, and stores current positive keys in the feature memory for future iterations. The key-storing mechanism results in a relatively smaller batch size requirement that is beneficial for reducing device memory and also makes it easy for feature sharing implementations for our proposed scheme. For the loss function, MoCo relies on InfoNCE loss Oord et al. (2018) as the contrastive mechanism to update its model parameters:

$$\mathcal{L}_{Q,K,N} = -log \frac{exp(Q \cdot K^+/\tau)}{exp(Q \cdot K^+/\tau) + \sum_{N \in M} exp(Q \cdot N/\tau)} \tag{1}$$

where query key $Q$ and positive key $K^+$ are the output vectors of server-side momentum model and the momentum model, respectively, obtained by processing two augmented views of the image. $N$ denotes all negative keys in the feature memory of size $M$. Importantly, the success of MoCo scheme highly depends on the "hardness" of its negative keys (Kalantidis et al., 2020; Robinson et al., 2020). The "hardness" of a negative key $N$ in the feature memory bank, can be determined by the similarity (inner-product) between $Q_t$ (at step $t$) and $N$; the smaller the similarity, the better the "hardness". We notice the "hardness" of negative key $N$ reduces quickly because model updates are in the direction of minimizing the InfoNCE loss in Eq. (1). As a result, MoCo adopts a slow-changing momentum model to produce consistent negative keys to add to the feature memory at the end of every training step and thereby maintains their hardness.

### 2.2 SPLIT FEDERATED LEARNING

Split Federated Learning (SFL) Thapa et al. (2020) is a recent collaborative learning scheme that focuses on high computation efficiency at the client side. It splits the original model architecture into two parts, the client-side model that contains all layers upto the "cut-layer" and the server-side model that contains the remaining layers. We distribute copies of client-side model $C_i$ to client-$i$'s local devices and instantiate the server-side model $S$ in a cloud server. To complete each training step, clients need to send the latent vectors (the output of client-side model) to the server, and the server processes latent vectors, computes the loss, performs backward propagation and returns the corresponding gradients to clients. Thapa et al. (2020) present two possible ways for server to process latent vectors sent by clients. In this paper, we use SFL-V1 where the server concatenates all clients' latent vectors and processes them altogether, which makes the equivalent batch size larger at the server and benefits contrastive learning. In contrast, in SFL-V2, client's latent vectors are processed sequentially in a first-come-first-serve manner and thus does not benefit from the "large batch". We provide details of SFL-V1 in Appendix A.2.

## 3 MOTIVATION

As mentioned in Section 1, the two challenges in extending FL-SSL to cross-client applications are high computing resource requirement and large data requirement.

### 3.1 HIGH COMPUTING RESOURCE REQUIREMENT

The first challenge is the computing resource requirement of training an SSL model locally. Using a compact backbone model may be accurate for supervised learning, but is not suitable for SSL as it has a much higher requirement on the model capacity. (Shi et al., 2021; Fang et al., 2021) show

that compact architectures like Mobilenet-V3 (Howard et al., 2019), EfficientNet Tan and Le (2019) suffer from over 10% accuracy degradation compared to a larger ResNet-18 architecture, while an even larger ResNet-50 model has over 15% better accuracy compared to ResNet-18 on ImageNet dataset (Deng et al., 2009). This means memory requirement for training an SSL model with high accuracy is very high. Using a smaller batch size reduces accuracy dramatically, as shown in Fig. 1(a), and is thus not an option. A FL-SSL scheme (ResNet-18 with a batch size of 128) costs 590.6 MFLOPs per image and over 1100MB of memory per client, which is not practical.

## 3.2 Large data requirement

The other major difficulty for FL-SSL to generalize to the cross-client case is the large data requirement. For cross-client applications, the amount of data available to each client can be very limited. For example, in a cross-silo medical application, a client can be a hospital with tons of data. In comparison, in a cross-client application, a client can be a patient who has limited amount of data.

The root of the problem lies in the difficulty to find hard negative samples when clients do not have enough local data. When the amount of data is larger, the chance for hard negative samples to be present becomes much higher. As a result, existing FL-SSL can only be successful for cross-silo applications where clients have large amount of data and can perform effective contrastive learning locally. As demonstrated in Fig. 1(b), we observe high accuracy when clients have 10K samples of data, while the accuracy drops quickly to around 30% when clients have only 500 samples.

Unfortunately, remote feature sharing Zhang et al. (2020); Wu et al. (2021; 2022d) in FL-SSL schemes cannot solve the large data requirement. These schemes update the shared feature memory less frequently because of significant communication overhead with each update. Since clients must synchronize their local copies of the shared feature memory each time a minor change happens, in Zhang et al. (2020); Wu et al. (2022d), new negative keys are added to the feature memory only once per epoch. Even so, the total communication overhead of remote feature sharing scales quadratically with the number of clients makes it not practical for cross-client case.

## 4 Method

### 4.1 Proposed MocoSFL

Our proposed MocoSFL is an innovative combination of SFL-V1 and MoCo-V2 (Chen et al., 2020b) as shown in Fig. 2. There are three key components. **First**, in each training step, the latent vectors sent by all clients are concatenated before being processed by the server-side model. This helps achieve a large equivalent batch size in order to support mini-batch training. **Second**, we use a shared feature memory which is updated by positive keys contributed by all clients in every training step. **Third**, we improve the non-IID performance by using a higher synchronization frequency.

Next, we will elaborate on how these three components in the proposed MocoSFL address the two challenges in Section 4.2 and Section 4.3. Section 4.4 describes how MocoSFL achieves better non-IID performance and Section 4.5 addresses the privacy and communication issues of the proposed scheme.

### 4.2 Reduce Hardware Resource Requirement

Choice of SFL helps reduce computational overhead and memory consumption at the client-end because of the much smaller client-side model. For example, on a CIFAR-10 ResNet-18 model with a batch size of 128, a client-side model with 3 layers only costs 13.7% of the FLOPs compared to the entire model, and its memory cost is 227MB, merely one fourth of the entire model. Furthermore, we reduce the batch size to 1 (also known as "micro-batch"), to further reduce the memory consumption. The use of micro-batch in local model training is only possible thanks to the latent vector concatenation mechanism which basically aggregates latent vectors sent by all clients into a big batch before sending it to the server. In addition, in a micro-batch setting, we replace the batch normalization layer by group normalization Wu and He (2018) and weight standardization (Qiao et al., 2019) to gain better accuracy performance. In Fig. 3(b), we compare the computation and memory consumption of the proposed MocoSFL with the FL-SSL scheme. MocoSFL with cut-layer of 3 achieves $\sim 288\times$

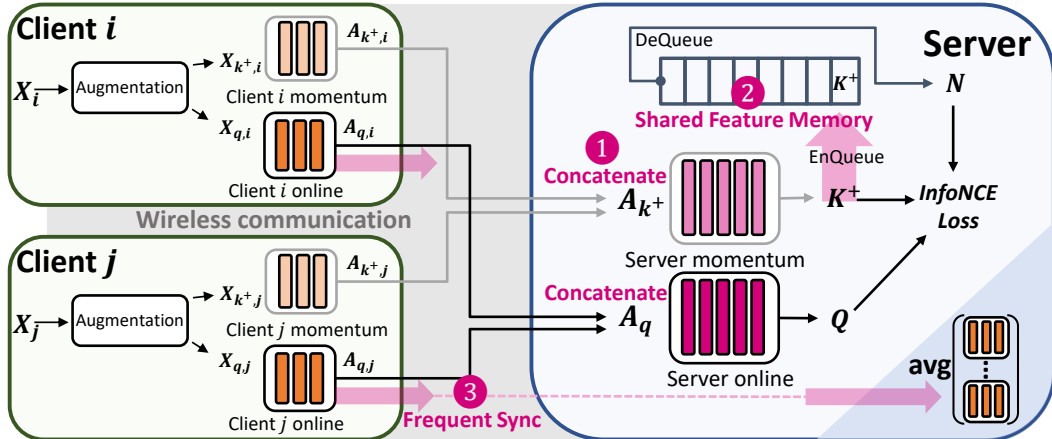

Figure 2: MocoSFL scheme. Three highlighted components are (1) latent vector concatenation, (2) shared feature memory, and (3) frequent synchronization.

reduction in memory consumption than FL-SSL and has 2%-10% higher accuracy. Details of accuracy evaluation are included in Section 5.1.

### 4.3 MITIGATE LARGE DATA REQUIREMENT

As indicated by (Kalantidis et al., 2020; Robinson et al., 2020), the "hardness" of a negative key $N$ heavily depends on its similarity with the current query key $Q$, given that $N$ and $Q$ have different ground-truth labels. To evaluate the hardness of negative key $N_0$ residing in the feature memory, we use the similarity measure (inner-product) between $N_0$ and $Q_t$, a freshly calculated query key at time $t$. In FL-SSL with feature sharing, the negative key is only updated after a long period of time to reduce communication overhead. As a result, the hardness diminishes quickly. In contrast, MocoSFL frequently updates its feature memory to maintain a good hardness. At every training step, a freshly calculated positive key $K_+$ is added to the tail of the queue, and the oldest one is popped out.

However, frequent updates of feature memory is not enough to ensure a high level of hardness. We also find it is necessary to use a large batch size and a large feature memory. This finding agrees with the study in Bulat et al. (2021) and also explains the accuracy drop for a small batch size in FL-MocoV2 in Fig. 1(a). To illustrate this, we consider the total similarity measure at time $t$ and make the following assumptions: For the newest batch of negative keys $N_t = K_{t-1}$ of size $B$ in the feature memory at time $t$, we assume the similarity measure between $N_t$ and $Q_t$ is a constant $\eta$ for all $t$. We also assume similarity of every batch of negative keys in feature memory gets reduced by a constant factor $\gamma$ ($\gamma < 1$) after each model update to represent the degradation caused by model updates. Thus, for a freshly computed query $Q_t$, its total similarity measure with negative keys in the feature memory can be represented as:

$$hardness = B\eta\gamma + B\eta\gamma^2 + ... + B\eta\gamma^{\lfloor M/B \rfloor} \tag{2}$$

$$= B\eta\gamma \times (\frac{1 - \gamma^{\lfloor M/B \rfloor}}{1 - \gamma}) \tag{3}$$

where $B$ is the batch size and $M$ is the feature memory size. We see that using a large batch size $B$ is beneficial as it helps bring more fresh negative keys and maintain better hardness. Also, using a larger feature memory (increasing $M$) can keep enough negative keys and contribute to a better total hardness. In the cross-client case, FL-SSL schemes can hardly meet these two requirements because of the small clients' memory. Nonetheless, MocoSFL can easily fulfill them because (1) latent vector concatenation enables a large equivalent batch size, and (2) feature memory hosted by the cloud server can be much larger.

### 4.4 IMPROVING NON-IID PERFORMANCE

We found that use of SFL results in fewer model parameters at the client side and hence smaller model divergence. Furthermore, introducing frequent synchronization in MocoSFL provides additional

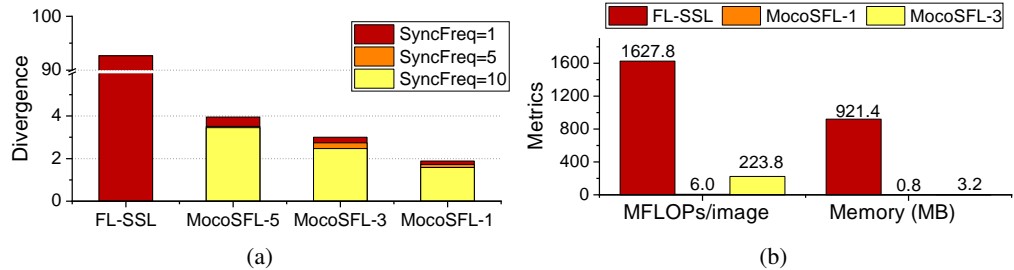

Figure 3: (a) Proposed MocoSFL reduces model divergence. (b) Computation overhead (FLOP counts of inference per image of the online model at client-end) comparison between FL-SSL scheme and MocoSFL schemes. MocoSFL-$L$: client-side model has $L$ layers.

reduction in model divergence and greatly improves the non-IID performance. According to Zhang et al. (2020); Zhuang et al. (2021; 2022), the model divergence between two models is calculated as the L2 norm of the weight difference. Following the same idea, the total divergence measure of a cross-client system can be measured as the average weight divergence of local models w.r.t. the global model during training:

$$divergence = \frac{1}{EN_C} \sum_{e=1}^{E} \sum_{i=1}^{N_C} \sum_{l=1}^{L} ||W_{e,l}^i - W_{e,l}^*||_2 \tag{4}$$

where $L$ denotes the number of layers in the client-side model, $E$ denotes the total number of synchronizations, $N_C$ denotes the number of clients, and $l, e, i$ are the respective indices for $L, E, N_C$. $W^*$ is the average of all client models $W^i$. MocoSFL reduces the model divergence with two orthogonal mechanisms. The first mechanism is the reduction of client-side model size, which directly results in a lower model divergence. As shown in Fig. 3(a), compared to FL-SSL scheme, MocoSFL has a much lower model divergence when the client-side model has less than 5 layers. The other mechanism is frequent model synchronization which helps reduce the model divergence. This is only possible in SFL because of communication overhead of sending weights in a smaller client-side model, is smaller. Fig. 3(a) also illustrates how model divergence further reduces as we increase the synchronization frequency.

## 4.5 IMPROVING PRIVACY AND COMMUNICATION OVERHEAD OF MOCOSFL

The proposed MocoSFL scheme is based on SFL and suffers from two issues – high overall communication overhead due to transmitting and receiving latent vectors and vulnerability to Model Inversion Attack (MIA) Fredrikson et al. (2015), in which the server can reconstruct clients' raw inputs from latent vectors, making clients' data privacy questionable. (We leave the details of its threat model and working mechanism in Appendix A.3) To address the privacy and communication issues in MocoSFL, we propose Target-Aware-ResSFL (TAResSFL). TAResSFL extends ResSFL Li et al. (2022) for self-supervised learning through: (1) target-data-aware self-supervised pre-training, and (2) freezing feature extractor during SFL training. TAResSFL also utilizes the bottleneck layer design from ResSFL to reduce the communication overhead.

In ResSFL Li et al. (2022), the server performs pretraining to build up the resistance to MIA using data from a different domain since it does not have access to clients' data. Next, the pretrained resistant client-side model is transferred to the clients and gets fine-tuned using SFL. TAResSFL improves the pretraining step by assuming that the server can get access to a small subset (<1%) of training data, together with large amount of data from another domain, and perform pre-training using self-supervised learning. Such a pretrained client-side model has better transferability, and can thus stay frozen during SFL process, thereby avoiding the expensive fine-tuning. As shown in Fig. 4, we blend the source dataset $X_s$ with a tiny subset of target dataset $X_t$ during the feature extractor training. The attacker-aware training has the min-max form:

$$\min_{\boldsymbol{W}_C, \boldsymbol{W}_S} \max_{\boldsymbol{W}_G} \underbrace{\mathcal{L}(S(\boldsymbol{W}_S; C(\boldsymbol{W}_C; [\boldsymbol{x}_q, \boldsymbol{x}_{k+}])))}_{\text{Contrastive Loss}} + \lambda \underbrace{\mathbb{R}(G(\boldsymbol{W}_G; C(\boldsymbol{W}_C; \boldsymbol{x}_q)), \boldsymbol{x}_q)}_{\text{Inversion Score}} \tag{5}$$

where $\mathbb{R}$ in the inner maximization denotes a similarity measure, for which we use the structural similarity index (SSIM) score Zhao et al. (2016). The inner maximization is used to train the simulated attack model $G$, whose function is reconstructing the activation back to a raw input state that is similar with ground-truth $X_q$. The outer minimization step goes in the direction of lowering contrastive loss, where the regularization term makes the model accurate as well as resistant to attack. These two steps are done alternatively to make the feature extractor resistant, and also be able to achieve good accuracy on the target dataset.

Fig. 4 presents the Target-aware ResSFL scheme. We use the resistant feature extractor to initialize client-side models during transfer step, as shown by the pink arrow in Fig. 4. Unlike ResSFL, here we freeze its parameter completely to maintain the resistance since any parameter change can cause resistance drop. However, freezing brings a noticeable accuracy drop of larger than 3%, even with the use of CIFAR-100 as source dataset if we perform pre-training by only using the source dataset (see Table 4). But if we blend the source data with a small portion of target data during pre-training, the accuracy can be greatly improved. Since the model accuracy validation is done by the server for monitoring purpose (Bhagoji et al., 2019), we believe that the server can separate out a small proportion of validation data to meet the target data availability assumption.

The freezing also greatly benefits hardware resource requirement since: (1) clients only need to transmit the latent vectors to the server and do not need to perform backward propagation using gradients from the server; (2) client-side model synchronization is not needed. As a result, MocoSFL with TAResSFL component achieves a $\sim 128\times$ (=5001.4/39.1) overall communication reduction compared to the original MocoSFL, as shown in Table 1. Here MocoSFL methods undergo 200 model synchronizations for 200 epochs while FL-SSL methods need 100 synchronizations (using the same setting in Zhuang et al. (2022)).

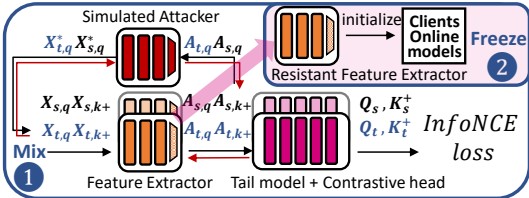

| Communication Overhead | Weights (MB) | Latent Vectors (MB) | Total (Relative (MB) Ratio) |
|---|---|---|---|
| FL-SSL methods Zhuang et al. (2022) | 8269.2 | 0.0 | 8269.2 (1.000x) |
| MocoSFL-1 | 1.4 | 5000 | 5001.4 (0.605x) |
| MocoSFL-3 | 57.9 | 5000 | 5057.9 (0.612x) |
| MocoSFL-1+TAResSFL | 0.0 | 39.1 | 39.1 (0.005x) |
| MocoSFL-3+TAResSFL | 0.0 | 39.1 | 39.1 (0.005x) |

Figure 4: Target-aware ResSFL scheme: (1) Target-domain data is used in pretraining; (2) Client-side model is frozen during training.

Table 1: Communication overhead per client. FL-SSL: 100 times of synchronization; MocoSFL: 200 times of synchronization.

## 5 EXPERIMENTAL RESULT

**Experimental Setting.** We simulate the multi-client MocoSFL scheme on a Linux machine, where we use different CPU threads to simulate different clients and a single RTX-3090 GPU to simulate the cloud server. We use ResNet-18 (He et al., 2016) for the majority of the experiments to better compare with existing SoTA (Zhuang et al., 2022). We use CIFAR-10 as the main dataset and also present results on CIFAR-100 and ImageNet 12-class subset as in Li et al. (2021). For the IID case, we assume the entire dataset is divided randomly and equally among all clients. For non-IID experiments, we mainly consider the pathological (aka. class-wise) non-IID distribution as in McMahan et al. (2017); Zhuang et al. (2022) where we assign 2 classes of CIFAR-10/ImageNet-12 data or 20 classes of CIFAR-100 data randomly to each client. We perform MocoSFL training for a total of 200 epochs, using SGD as the optimizer with an initial learning rate of 0.06. For accuracy performance evaluation, we adopt similar linear probe methods as in Grill et al. (2020); Zhuang et al. (2022), where we train a new linear classifier on the outputs generated by the MocoSFL backbone model. We include details of hyper-parameter choices and evaluations in Appendix A.1.

### 5.1 ACCURACY PERFORMANCE

**Improved non-IID performance.** Fig. 5 shows how the increased synchronization frequency can significantly improve the non-IID accuracy. We present results for the cut-layer choices of 1 and 3 convolutional layers in the client-side model, represented by "MocoSFL-1" and "MocoSFL-3", respectively. We attribute the improved accuracy to the reduction in model divergence.

**Comparison with FL-SSL.** When synchronization frequency of the MocoSFL is set to 10 (per epoch) for the 5-client cases, MocoSFL achieves significantly better non-IID accuracy performance than Zhuang et al. (2022) on CIFAR-10 dataset due to lower model divergence (see Table 2). On CIFAR-100 dataset, with $N_C = 5$, we observe our method has lower accuracy than FL-SSL methods. We hypothesize that the performance of MocoSFL is more sensitive to the model complexity, and thus has limited performance for a more complex task like CIFAR-100. When model complexity is high enough, for instance, on a larger ResNet-50 model, accuracy of our method is ~4% higher than Zhuang et al. (2022) as shown in Appendix B.7. Furthermore, our methods outperform FL-SSL methods by a large margin in 20-client cases thanks to the feature sharing aspect.

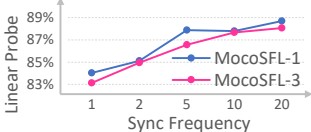

Figure 5: Effect of increasing synchronization

Table 2: Non-IID performance comparison (linear probe accuracy)

| Method | CIFAR-10 | | CIFAR-100 | |
|---|---|---|---|---|
| | $N_C = 5$ | $N_C = 20$ | $N_C = 5$ | $N_C = 20$ |
| FL-BYOL (Zhuang et al., 2022) | 83.34 | 75.77 | 61.78 | 52.78 |
| MocoSFL-1 (ours) | 87.81 | 85.84 | 58.78 | 57.80 |
| MocoSFL-3 (ours) | 87.29 | 85.32 | 57.70 | 57.52 |

**Cross-client Performance.** Our proposed MocoSFL can generalize from a cross-silo application (with upto 20 clients) to a cross-client application with 100, 200, and 1000 clients. Note that none of the previous FL-SSL methods can scale to such a large number of clients. For the hyper-parameter choices, we follow two principles introduced in Appendix A.1 – we let each client use a batch size of 1 and use the synchronization frequency of $f_S = (1000/N_C)$/epoch, and we set the client sampling ratio to $100/N_C$ to keep the same equivalent batch size at the server end. The results are shown in Table 3. Note that each client has only 50 data samples in the 1000-client case. MocoSFL's accuracy for IID case is high when $N_C$ increases from 100 to 1,000, though its accuracy drop by 1% for non-IID case. This small drop is because model divergence scales with number of clients as described in Section 4.4.

Table 3: MocoSFL cross-client accuracy performance (linear probe accuracy) of ResNet-18 model on CIFAR-10, CIFAR-100 and Imagenet-12 datasets with different number of clients $N_C$.

| Method | Dataset | IID | | | non-IID | | |
|---|---|---|---|---|---|---|---|
| | | $N_C = 100$ | $N_C = 200$ | $N_C = 1000$ | $N_C = 100$ | $N_C = 200$ | $N_C = 1000$ |
| MocoSFL-1 | CIFAR-10 | 87.29 | 87.38 | 87.51 | 87.71 | 87.39 | 86.46 |
| | CIFAR-100 | 58.91 | 59.15 | 58.85 | 59.22 | 58.90 | 56.75 |
| | ImageNet-12 | 92.02 | 91.73 | 91.76 | 92.24 | 91.44 | 91.28 |
| MocoSFL-3 | CIFAR-10 | 87.29 | 87.15 | 87.25 | 87.10 | 85.22 | 84.75 |
| | CIFAR-100 | 58.41 | 58.30 | 58.80 | 58.69 | 58.59 | 56.88 |
| | ImageNet-12 | 92.08 | 92.24 | 92.02 | 92.60 | 91.83 | 91.28 |

## 5.2 PRIVACY EVALUATION

| Method | Metric | Target Data | | |
|---|---|---|---|---|
| | | 0.0% | 0.5% | 1.0% |
| **MocoSFL-1** | Accuracy (%) | 81.14±0.47 | 80.78±1.34 | 79.96±2.96 |
| | Attack MSE | 0.039±0.005 | 0.033±0.014 | 0.039±0.002 |
| **MocoSFL-3** | Accuracy (%) | 81.19±2.32 | 80.51±1.49 | **83.13±2.40** |
| | Attack MSE | 0.045±0.003 | 0.035±0.003 | **0.039±0.002** |

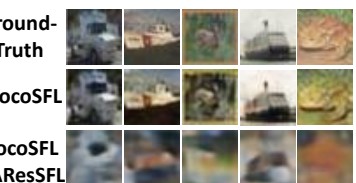

Table 4: Accuracy and MIA resistance performance (Attack MSE) of MocoSFL+TAResSFL. (Averaging 3 random seeds)

Figure 6: Visualization of MIA reconstructed images using TAResSFL.

We use $G$ to perform MIA attack; the architecture information of $G$ is given in Appendix A.3. We assume the real-time attacker also uses an attack model with the same architecture as $G$. For the pre-training step, we use a fixed hyper-parameter choice of $\lambda = 2.0$ and a target SSIM level of 0.6 to limit the strength of regularization. We assume 0.0%, 0.5% and 1.0% of the target dataset CIFAR-10 is accessible, and we use CIFAR-100 dataset as source dataset to assist the pre-training. We set the cut-layer choices to 1 and 3 and set the #clients to 100 for the training process.

**Successful mitigation of MIA.** As shown in Table 4, applying TAResSFL can achieve good accuracy performance (>81%) as well as high enough MIA resistance (>0.020) for most cases. Fig. 6 shows the visualization of MocoSFL-3 when 1.0% target data is available. We observe the reconstructed images are much more noisy and blurry such that the subject can be successfully hidden.

**Larger cut-layer allows a better resistance-accuracy tradeoff.** Using a smaller cut-layer of 1 seems a better choice in terms of accuracy and hardware requirement. However, as shown in Table 4, the accuracy and resistance tradeoff seems much better by using a cut-layer of 3. We believe that the extra client-side model complexity helps in the optimization of both accuracy and resistance objectives. So, applying TAResSFL with a larger cut-layer is a more favorable option.

## 5.3 Hardware Demonstration

Finally, we compare the total hardware resource cost of the proposed MocoSFL and "Mo-coSFL+TAResSFL" with synchronization frequency of 1/epoch for 200 epochs, and FL-SSL (Zhuang et al., 2022) with 500 local epochs per client and synchronization frequency set to 1 per 5 local epochs (original setting). For MocoSFL, we use 1,000 clients with batch size of 1, and cut-layer of 3. For FL-SSL, to achieve similar accuracy, we use 5 clients with batch size of 128. And we assume the data follows the default 2-class non-IID setting. To evaluate overhead, we use commonly-used libraries such as 'fvcore' for FLOPs and 'torch.cuda.memory_allocated' for memory (please see Appendix A.5 for detail.) For power measurement, we use a Raspberry Pi 4B equipped with 1GB memory as one real client and simulate all other clients on the PC.

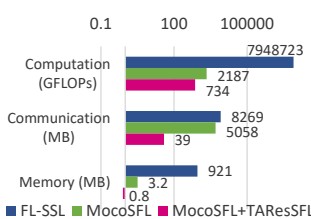

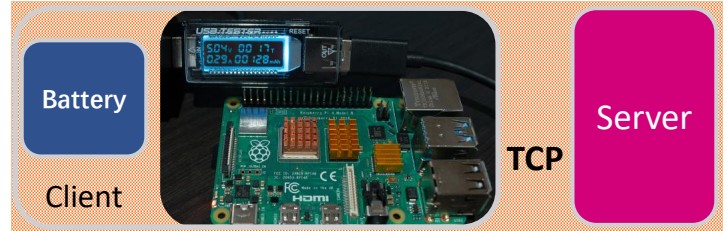

Figure 7: Hardware resource comparison between FL-SSL and MocoSFL schemes.

Figure 8: Raspberry Pi setup. The Raspberry connects to the output via a measurement tool, and communicates wirelessly with the server using websocket (TCP).

As shown in Fig. 7, FL-SSL requires 7,949 TFLOPs for the entire training session as each client needs to perform 500 local epochs on 10K data and a memory consumption of 921 MB. The communication overhead due to synchronization has a high cost of around 8,269 MB. Thus, FL-SSL is only suitable for cross-silo situation. For "MocoSFL+TAResSFL", hardware requirements are reasonable since computation is only 734 GFLOPs and communication is around 39 MB in total. The memory consumption is a tiny 0.8 MB mostly due to the weight parameters since TAResSFL does not require local training. As shown in Fig. 8, our measurement using a USB multimeter shows the proposed MocoSFL running on the Raspberry Pi only draws power of 2.26W, in average, and consumes around 9,100 mAh on a 5V battery.

## 6 Conclusion

We propose MocoSFL, a collaborative SSL framework based on SFL. The proposed framework addresses hardware resource requirement at client-side by enabling small batch size training and computation offloading. It also relieves the large data requirement of local contrastive learning by enabling effective feature sharing. The proposed scheme is the only one that can support a large number of clients. In combination with a ResSFL-based module, it addresses privacy concerns of MocoSFL. Finally, it achieves even better IID/non-IID performance with much lower hardware requirement than the SoTA FL-based SSL methods.

## 7 ETHICS STATEMENT

In this work, we address two practical issues of hardware (compute and memory) resources and large data requirement in collaborative SSL by using a SFL-based scheme. Compared to the conventional approaches, the resulting MocoSFL scheme has significant advantages in affordability and is much easier to deploy. Apart from being environmental-friendly, MocoSFL makes it possible to reach more clients, especially for those in poverty or under-represented minorities, and thus eliminate potential human-bias and unfairness issues of the ML service.

We address the privacy issues in the proposed MocoSFL by proposing TAResSFL module to use in sensitive applications. Fig. 6 shows that the subject of the raw image can be successfully hidden.

**More discussion on the privacy of "MocoSFL+TAResSFL".** We notice a line of works known as instance encoding (Huang et al., 2020; Yala et al., 2021), which try to protect users' data by transforming the original dataset to a distorted version such that they cannot be recognized by humans while an arbitrary deep learning model can still learn useful information from it (i.e. achieve high accuracy on a classification task). MocoSFL, especially with the TAResSFL, where the client-side model is frozen, have some similarities since the frozen client-side model can be seen as a transformation, and the collection of latent vectors can be regarded as the transformed dataset. However, "MocoSFL+TAResSFL" scheme has two favorable properties that make it distinct from instance encoding methods. **Domain dependency.** First, SFL only finds a transformation method for a given task. Since the transformation method itself (the client-side model) heavily depends on the target domain information (as we need access to the target domain data to train the client-side model), it cannot work on data from another domain. As the output of the "client-side model transformation", latent vectors are only useful for current problem without any transferability guarantee. However, instance encoding methods intend to derive a general transformation method that can work across domains, targeting a harder problem. **Risk Control.** Instance encoding methods publish the transformed dataset to the wide public which is risky. As pointed out by Carlini et al. (2021): all the raw data will be leaked if a successful decryption method is invented in the future even if it does not exist now. While in our proposed scheme, the latent vectors from clients will only be accessible to the server party, thereby reducing the risk. Since we already provide TAResSFL to mitigate the MIA attack that can be possibly launched from the honest-but-curious server, the risk is minimized. To eliminate possibility of future advanced attack, we can introduce a protocol that requests the server party to regularly delete[1] these latent vectors immediately after the training is done.

## 8 REPRODUCIBILITY STATEMENT

To make it easier for readers to reproduce the results in this work, we open-source our code at `https://github.com/SonyAI/MocoSFL`. We also provide detailed explanation on the MocoSFL training and evaluation hyperparameters, collaborative learning system hyperparameters in Appendix A.1, and TAResSFL module hyperparameters in Appendix A.4.

## 9 ACKNOWLEDGMENT

This work is funded by Sony AI.

---

[1]This is a common practice according to https://oag.ca.gov/privacy/ccpa

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
