# OpenReview forum: "MocoSFL: enabling cross-client collaborative self-supervised learning"
_ICLR.cc/2023/Conference — ICLR 2023 notable top 5%_

### Official Review · Reviewer_zHbH · 2022-10-22

**Confidence:** 4
**Correctness:** 4
**Technical Novelty And Significance:** 4
**Empirical Novelty And Significance:** 4
**Recommendation:** 8

**Clarity, Quality, Novelty And Reproducibility:**

Clarity: The limitations of current FL-SSL methods are clearly identified. The proposed MocoSFL is clearly explained.

Quality: The whole paper is well structured and well written. The experimental results are extensive and convincing.

Novelty: To my knowledge, the proposed MocoSFL and the combination with TAResSFL is novel. Also, seems MocoSFL is the first work that can deal with as large as 1,000 clients in federated self-supervised learning. Hence, I think this paper could be a milestone work in advancing the area of federated self-supervised learning.

Reproducibility: Detailed experiment settings in the paper are sufficient to reproduce the results. Authors also plan to release code upon acceptance in their REPRODUCIBILITY STATEMENT.


**Strength And Weaknesses:**

Strengths

1. The whole paper is very well structured and well written.
2. The proposed MocoSFL can be scaled to a large number of clients (1,000 clients for the first time).
3. MocoSFL seems to be a very practical framework, which requires less memory, less communication, and consumes less power.
4. Authors also test the effectivenes of MocoSFL on a real Raspberry Pi 4B device, which validates its potential to land in industry.
5. The experimental design is solid and the results are convincing. All the key factors in a system are well considered and addressed, including model performance, memory usage, computation cost, communication cost, and privacy.

Weaknesses

1. All the chosen datasets are CV centered, I want to know whether MocoSFL can be extended to the other types of data. More discussion is expected.
2. Seems FL-BYOL is the only baseline. Are there any potential baseline? Could you elaborate more on this?
3. Figure 5 can be better illustrated with x-axis and y-axis lines with arrows.


**Summary Of The Paper:**

This paper proposes MocoSFL, a collaborative SSL framework based on SFL. The proposed framework addresses hardware resource requirement at client-side by enabling small batch size training and computation offloading. It also relieves the large data requirement of local contrastive learning by enabling effective feature sharing. To address the privacy issue and communication overhead of MocoSFL, this paper further introduces the TAResSFL module. Combined with TAResSFL, MocoSFL can support 1,000 clients for the first time, achieve better non-IID performance, and largely reducte memory usage, computation and communication cost.

**Summary Of The Review:**

FL under self-supervised learning setting is still a very new but very practical problem. This paper clearly identifies the major challenges in deriving high accuracy in FL-SSL schemes for cross-client applications, and proposes a novel collaborative SSL framework called MocoSFL based on Split Federated Learning (SFL) and Momentum Contrast (MoCo). MocoSFL is the first work that can deal with as large as 1,000 clients in federated self-supervised learning, and tests its effectivenes on a real Raspberry Pi 4B device. Overall, this paper is very interesting and to my knowledge novel. This paper could be a milestone work in advancing federated self-supervised learning. Hence, I would like to vote for acceptance.

---

> ### Author Response · Authors · 2022-11-17
> **Response to Reviewer zHbH**
>
> Thanks for your valuable comments. We hope our response below can adequately address your concerns.
>
> **Q1:** All the chosen datasets are CV centered, I want to know whether MocoSFL can be extended to the other types of data. More discussion is expected.
>
> **Answer:** Our paper focuses on CV classification tasks to demonstrate the potential of the proposed method. We will consider generalization to other application domains such as Text and Voice in our future work.
>
> **Q2:** Seems FL-BYOL is the only baseline. Are there any potential baseline? Could you elaborate more on this?
>
> **Answer:** We only list FL-BYOL in the main paper because it is the best among all techniques presented in FedEMA. We provide results for other schemes such as FedU, FedCA in Appendix B.7.
>
> **Q3:** Figure 5 can be better illustrated with x-axis and y-axis lines with arrows.
>
> **Answer:** Thanks for the suggestion. We remade the figure with the suggested changes in the revised version.

---

### Official Review · Reviewer_Ddhx · 2022-10-23

**Confidence:** 5
**Correctness:** 4
**Technical Novelty And Significance:** 3
**Empirical Novelty And Significance:** 4
**Recommendation:** 8

**Clarity, Quality, Novelty And Reproducibility:**

The paper clearly explains the problems and presents a novel combination of SFL and MoCo as solutions. The paper is well-written and organized. Moreover, the authors provide detailed experiment settings in appendices and promise the future release of code repository.

**Strength And Weaknesses:**

Strengths
(1)	This paper firstly identifies two practical limitations of FL-SSL methods that make them unable to support cross-client applications.

(2)	The roots of these two challenges are discussed well in Section 2.

(3)	This paper proposes a novel combination of SFL and MoCo and utilizes three components to get the best of the two. The resulting MocoSFL enables a small client-side model, a large equivalent batch size, and effective feature sharing hence mitigates two challenges.

(4)	The paper presents good reasoning on why the proposed MocoSFL is successful in mitigating large data requirements and improving non-IID performance.

(5)	Extensive experimental results demonstrate the advantages of the proposed method compared to state-of-the-art methods.

(6)	The paper honestly points out the remaining privacy and communication issues of MocoSFL, and correspondingly presents TAResSFL module to mitigate these issues.

(7)	At the end, the paper presents a hardware demo to showcase the practicality of the proposed method, showing that clients with a cheap device can participate in federated self-supervised learning without any problem.

Weaknesses

(1)	Experiment shows up to 1,000 clients. It would be interesting to see the accuracy performance when the number of clients is greater than 1,000.

(2)	MocoSFL results are limited - only for ResNet architectures. How it performs on other models?

(3)	In TAResSFL, the availability of a close-related source domain dataset and a small portion of target domain data is questionable and should be better justified.

(4)	Hyper-parameter choices of TAResSFL seems not so easy. Could you give more hints on how to wisely or automaticly choose hyper-parameters?

(5)	(minor) The privacy evaluation only uses mean square error (MSE) as measurement, did you ever consider other metrics such as structural similarity index measure (SSIM), Peak signal-to-noise ratio (PSNR)?


**Summary Of The Paper:**

This paper introduces a federated self-supervised learning method named MocoSFL. based on Split Federated Learning (SFL) and MoCo.

By leveraging a small client-side model, vector concatenation, and effective feature sharing, MocoSFL solves two challenges: high computing resources and large data requirements. Thus, MocoSFL makes it practical for a cross-client application which typically has a very large number (>100x) of clients, and each client holds a small amount of data and has limited computing capability.

In addition, this paper introduces the TAResSFL module to effectively address the privacy threat and communication overhead of MocoSFL. MocoSFL with TAResSFL module can support 1,000 clients for the first time, and achieve over 1000x reduction in memory usage, 10000x reduction in computation, and 200x reduction in communication overhead. Moreover, compared to state-of-the-art methods, MocoSFL achieves better non-IID accuracy because of its small local model divergence.


**Summary Of The Review:**

This seems like a particularly interesting paper. This paper identifies practical problems in federated self-supervised learning and provides a novel MocoSFL framework to effectively solve them. The paper presents an in-depth discussion of both the problems and the solutions and shows extensive experimental results to support the arguments.  Finally, with the remaining issues addressed, the hardware demo leverages the importance and practicality of this work.

---

> ### Author Response · Authors · 2022-11-17
> **Response to Reviewer Ddhx**
>
> Thanks for your time reviewing our paper and the thoughtful comments. Following your suggestions, we have run additional experiments and added the new results to the revised paper. We hope the new version can adequately address your concerns.
>
> \
> **Q1:** Experiment shows up to 1,000 clients. It would be interesting to see the accuracy performance when the number of clients is greater than 1,000.
>
> **Answer:** Thanks for this insightful comment. Our experiments show at most 1,000 clients because CIFAR-10 dataset has only 50K data which translates to only 50 images for each client, which is already a ‘small data’ case. For MocoSFL, we swept the number of clients from 100 to 1000, and found only a very small accuracy degradation (less than 1%), which suggests further scaling should not be a problem. For MocoSFL with TAResSFL, the performance will not be affected by the number of clients since the client-side model is frozen, and having more clients is equivalent to having more “latent vectors” concatenated at the server-side model.
>
> \
> **Q2:** MocoSFL results are limited - only for ResNet architectures. How it performs on other models?
>
> **Answer:** Thank you for your valuable suggestion. In Appendix B. 2, We have included results for MocoSFL on MobilnetV2 and VGG-13 under IID and Non-IID settings under the 100-client setting. Compared with ResNet-18 and ResNet-50, their accuracy performance is not as good. For example, on Mobilenet-V2, its IID accuracy is 81.4% and Non-IID accuracy drops to 76.0%. As we expected, the poor performance is because of Mobilenet-V2's smaller model size compared to ResNet-18/50.
>
> \
> **Q3:** In TAResSFL, the availability of a close-related source domain dataset and a small portion of target domain data is questionable and should be better justified.
>
> **Answer:** The existence of target domain data can be justified by using a subset of validation data. We provide additional evidence to support our assumption that the server can have a subset of validation data. For instance, in Lin et al. [1], the early stopping strategy in ensemble distillation requires the server to have some validation data, in Nagalapatti et al. [2], the irrelevant client mitigation method  heavily relies on the server to have validation data, in Wang et al. [3] FL’s non-IID performance is improved using available validation data and in Zhang et al. [4] the block-chain-based FL system relies on validation data for for quality evaluation.
> However, we admit the assumption is not always practical and many existing works [5, 6] are opposed to such an assumption. According to our experiment results, even if 0% validation data is available, our TAResSFL algorithm still achieves > 80% accuracy.
>
> Reference
>
> [1] Lin et al. Ensemble Distillation for Robust Model Fusion in Federated Learning
>
> [2] Nagalapatti et al. Game of gradients: Mitigating irrelevant clients in federated learning
>
> [3] Wang et al. Optimizing Federated Learning on Non-IID Data with Reinforcement Learning
>
> [4] Zhang et al. Refiner: A reliable incentive-driven federated learning system powered by blockchain
>
> [5] Wu et al. Mitigating Backdoor Attacks in Federated Learning
>
> [6] Fung et al. Mitigating Sybils in Federated Learning Poisoning
>
> \
> **Q4:** Hyper-parameter choices of TAResSFL seems not so easy. Could you give more hints on how to wisely or automaticly choose hyper-parameters?
>
> **Answer:** Thanks for the insightful question. We do not have an easy way to choose the hyper-parameters – sorry! Our choice was based on intuition and trial-and-error. Please use the recommended hyper-parameter settings which we tested for multiple cases. All cases achieve comparatively good accuracy and resistance.
>
> \
> **Q5:** (minor) The privacy evaluation only uses mean square error (MSE) as measurement, did you ever consider other metrics such as structural similarity index measure (SSIM), Peak signal-to-noise ratio (PSNR)?
>
> **Answer:** Thanks for your suggestion. We provide SSIM and PSNR results for Table 4 in Appendix B. 5.

---

### Official Review · Reviewer_Sd6w · 2022-10-25

**Confidence:** 4
**Clarity, Quality, Novelty And Reproducibility:** 1) The overall structure of this pape…
**Correctness:** 4
**Technical Novelty And Significance:** 3
**Empirical Novelty And Significance:** 3
**Recommendation:** 8

**Strength And Weaknesses:**

Strength:

1) The practical value offered by MocoSFL is interesting: I think MocoSFL is a very practical solution that can support a large number of clients (1,000 clients) to conduct collaborative self-supervised learning (SSL).
2) The proposed MocoSFL has experienced enormous evaluations: The proposed MocoSFL takes into account both IID and non-IID settings, as well as the cross-client and cross-silo cases; experiments are carried out in both simulated and real hardware devices.
3) The proposed scheme takes into account not only model performance, hardware requirements, and communication overhead, but also privacy concerns.
4) Because most IoT devices have limited memory and most data is unlabeled in reality, the investigated problem is both timely and important to the community.
5) The paper's writing is good, and the overall structure is clear.


Weaknesses:

1)  In addition to model inversion attacks, I am wondering whether MocoSFL with the TAResSFL module has the potential to defend against other privacy attacks, such as the Membership Inference Attack.
2)  Although 1,000 clients are already a SOTA achievement, it would be great to also analyze how many clients can be supported by MocoSFL at most. Is there any turning point here?
3) I am very interested in the possibility of landing MocoSFL in ubiquitous IoT devices, so I want to know what is the min hardware requirement to implement MocoSFL, It is better to provide more discussions on this part?


**Summary Of The Paper:**

This paper aims to address the issues existing in current collaborative self-supervised learning (SSL) schemes and proposes MocoSFL, a collaborative SSL framework based on Split Federated Learning (SFL) and Momentum Contrast (MoCo). In MocoSFL, the large backbone model is split into a small client-side model and a large server-side model, and only the small client-side model is processed locally on the client’s local devices.
Specifically, there are three key components in MocoSFL: (i) vector concatenation which enables the use of small batch size and reduces computation and memory requirements by orders of magnitude; (ii) feature sharing that helps achieve high accuracy regardless of the quality and volume of local data; (iii) frequent synchronization that helps achieve better non-IID performance because of smaller local model divergence. Numerous evaluations are conducted to evaluate the effectiveness of the proposed MocoSFL.


**Summary Of The Review:**

The proposed MocoSFL takes into consideration not only model performance, hardware requirements, and communication overhead, but also privacy concerns. As a result, it would garner interest from a diverse community, including but not limited to FL, SSL, communication, privacy, and so on. MocoSFL also provides the industry with practical benefits by supporting a large number of clients and testing on real devices. This paper's overall structure is very clear, and the writing is also excellent. The paper as a whole is of high quality and of sufficient interest to the community.

---

> ### Author Response · Authors · 2022-11-17
> **Response to Reviewer Sd6w**
>
> Thank you very much for the positive feedback and valuable comments. We hope the following new results and clarifications can address your concerns.
>
> **Q1:** In addition to model inversion attacks, I am wondering whether MocoSFL with the TAResSFL module has the potential to defend against other privacy attacks, such as the Membership Inference Attack.
>
> **Answer:** Thanks for the valuable comments. The TAResSFL module is designed for addressing model inversion attacks, which reveal the raw images themselves and thus are stronger than membership inference attacks that only reveal whether a given sample is in the training set. However, this does not mean that TAResSFL can mitigate any weaker attack. Actually, membership inference attacks in a Federated setting have not been investigated thoroughly and we would like to investigate them in the future. Thank you for bringing this to our attention.
>
> **Q2:** Although 1,000 clients are already a SOTA achievement, it would be great to also analyze how many clients can be supported by MocoSFL at most. Is there any turning point here?
>
> **Answer:** Thanks for this insightful question. Our experiments show at most 1,000 clients because CIFAR-10 dataset has only 50K data which translates to only 50 images for each client, which is already a ‘small data’ case. For MocoSFL, we swept the number of clients from 100 to 1000, and found only a very small accuracy degradation (less than 1%), which suggests further scaling should not be a problem. For MocoSFL with TAResSFL, the performance will not be affected by the number of clients since the client-side model is frozen, and having more clients is equivalent to having more “latent vectors” concatenated at the server-side model.
>
> **Q3:** I am very interested in the possibility of landing MocoSFL in ubiquitous IoT devices, so I want to know what is the min hardware requirement to implement MocoSFL, It is better to provide more discussions on this part?
>
> **Answer:** Thanks for the suggestion. We listed the hardware resource consumption in Figure. 7 in the ideal case, for which we added a section in Appendix A. 5 to describe the calculation detail. However, in practice, memory consumption should also include the OS kernel and python library. In the Raspberry Pi 4B demonstration, MocoSFL consumes ~800MB of memory. But we project that with a specially crafted OS kernel and library support, the requirement in hardware can be further reduced.

---

### Official Review · Reviewer_bsEM · 2022-10-25

**Confidence:** 4
**Correctness:** 3
**Technical Novelty And Significance:** 2
**Empirical Novelty And Significance:** 2
**Recommendation:** 6

**Clarity, Quality, Novelty And Reproducibility:**

The proposed algorithm is clear and the paper is well-written, but the novelty is questionable.

**Strength And Weaknesses:**

Strengths: The chosen topic of collaborative SSL is an important and timely topic. The paper is clearly written. Performance is shown to be better than SOTA in some cases.

Weaknesses:

The novelty is limited as the work represents a combination of known methods. The design choices made during this combination all seem right but are also either already known or easily anticipated (like latent vector concatenation to create a large
batch size for the server-side model, sharing of features and more frequent synchronization). As for the target-aware variation, it is not clear whether the availability of local clients' training data (however small) is justified. Sure, as the authors mention, Bhagoji et al., 2019 considers the possibility of server being able to validate local models, but note that Bhagoji et al. mean to raise the bar for attackers (in an effort to prove that attacks are possible). In contrast, the present authors actually consider the possibility of using the local client data to guard against MIA attacks (and to reduce communication overhead following the feature extractor freezing). My impression is that this is a highly non-standard FL setting, and I would want to see more convincing references.

Specific questions/issues include:

The obvious baselines FedEMA and FedU are missing from Table 2 and Table 3.
From Table 5, the results for Non-IID cases with Nc=5 or Nc =100 are better than IID cases which is counter-intuitive. Could the authors provide justification for this performance gain?
Can the authors explain why the latent vector size in Table 1 changed from 5000 to 39.1 MB? Is it just because the backward gradients are not included?
In Figure 7, how the GFLOPs for computation are measured? Specifically, does it include the FLOPs needed for backpropagation?
The authors claim this technique to be competitive in cross-silo setting. However, the paper requires having batch size in range of 100-200 at the server-end which seems impractical considering there exists millions of samples/clients. Could the authors comment on this?
Eq (2) finds that having a large batch-size can maintain hardness of negative samples, which seems to contradicts the empirical results in Table 6 where increasing the batch size further to 400 degrades the performance. Reasons?
Did the authors implemented/reproduced the main results of baselines works e.g. FedEMA? It seems like the results are exactly as in the original paper.
How does this method compared against a FedMoco (Nanqing Dong, Irina Voiculescu: “Federated Contrastive Learning for Decentralized Unlabeled Medical Images”, 2021; arXiv:2109.07504) which also utilized MoCo in federated learning setup?
Fig 3 and Fig 7 contains computation results per clients only. How much computation burden is added to the server side?


**Summary Of The Paper:**

Collaborative self-supervised learning is considered based on a combination of known split federated learning and self-supervised contrastive leaning strategies. The former bases on model splitting between server and local clients while the latter depends on momentum-based key generation. The proposed combination, called MocoSFL, has some complexity/performance advantages over exiting FL-based solutions. A variation of this idea, based on pretraining (and later freezing) of feature extractors utilizing a small amount of clients' training data, is also proposed to improve communication overhead as well as resistance to MIA attacks.

**Summary Of The Review:**

The addressed topic - collaborative SSL - is important and timely. But the novelty here is limited as the work represents a combination of known methods. The design choices made during this combination all seem right but are also either already known or easily anticipated. As for the target-aware variation, the availability of local clients' training data at the server needs a stronger justification (is this a realistic/fair assumption? is it actually done in practice somewhere?). There are a number of questions/issues raised in the Weaknesses section above.

---

> ### Author Response · Authors · 2022-11-11
> **Response to Reviewer bsEM: (2/2)**
>
> Please first see "Response to Reviewer bsEM: (1/2)"
>
>
> \
> **II: Question on Hardware-related Results:**
>
> Sorry for missing details of the hardware evaluation. We have now included the details in Appendix A.5 section.
> **Q-II1:** Can the authors explain why the latent vector size in Table 1 changed from 5000 to 39.1 MB? Is it just because the backward gradients are not included?
>
> **Answer:** Thanks for the good question. Please refer to the newly added Appendix A.5 for the communication calculation detail. The latent vector has been reduced from 5000 to 39.1 MB because (1) the bottleneck layer (Conv layer with a channel size of 4 and stride of 2)  that we append to client-side model in TAResSFL compress the data, and (2) there are no backward gradients. **For example**, if the original feature size is 128x16x16, then after bottleneck layer, the feature size becomes 4x8x8, which is about a 64x reduction in communication. And because no backward gradients are needed, so we get a 128x reduction.
>
>
> \
> **Q-II2:** In Figure 7, how the GFLOPs for computation are measured? Specifically, does it include the FLOPs needed for backpropagation?
>
> **Answer:** Sorry for missing the detail. Please refer to the newly added Appendix A.5 for the computation calculation detail. And yes, backpropagation is included in the FLOPS count.
>
>
> \
> **Q-II3:**  Fig 3 and Fig 7 contains computation results per clients only. How much computation burden is added to the server side?
>
> **Answer:** The computation burden on the server side is similar to that of a centralized learning scheme. It is actually a little less because client-side model is computed at the client-end and not in the server. Simply put, the server processes a total of #Number of selected clients * # Number of data per client in each epoch for 200 epochs. We use a single 3090 GPU to handle the server-side computation.
>
> **III: Question on Practicality and Further Comparison:**
>
> \
> **Q-III1:** The authors claim this technique to be competitive in cross-silo setting. However, the paper requires having batch size in range of 100-200 at the server-end which seems impractical considering there exists millions of samples/clients. Could the authors comment on this?
>
> **Answer:** The key point in showing results for 100-1000 clients is to demonstrate that the proposed learning scheme can succeed even under small data. For example, having 1000 clients to process CIFAR-10 data, which consists of 50K data in total, means that each client only has 50 data. This is sufficient as a demonstration for “small data”.
> The only concern is the increasing batch size when more clients simultaneously participate. The server does not have infinite resources and so the batch size cannot be infinitely large. To mitigate this problem, we use client sampling (by setting a ratio < 1) to only allow a subset of clients to participate in each round to limit the batch size to a constant. Client sampling [1] is a standard approach used in FL schemes.
>
> **References:**
>
> [1] Charles et al. - On large-cohort training for federated learning.
>
> \
> **Q-III2:** How does this method compared against a FedMoco (Nanqing Dong, Irina Voiculescu: “Federated Contrastive Learning for Decentralized Unlabeled Medical Images”, 2021; arXiv:2109.07504) which also utilized MoCo in federated learning setup?
>
> **Answer:** Thanks for pointing out this important work. FedMoco is a good piece of work that demonstrates the success of FL + MoCo for medical application in a #client = 3, 6 setting. However, we did not consider FedMoco as an appropriate comparison candidate for the following reasons: (1) In cross-silo case, FedMoco only shows results for medical datasets and its code does not seem to be available. Also, the combination of FL and Moco has also been done in FedEMA paper, whose results are better than FedMoco. (2) In cross-client case, FedMoco may not work as it does not present any techniques to address small data and limited hardware resource problem.

---

> > ### Comment · Reviewer_bsEM · 2022-11-27
> > **thanks for the response**
> >
> > I thank the authors for their substantial efforts in answering the questions I raised. I feel my concerns are suuficiently addressed in most points. I particularly liked the comments made for Q-III1 and Q-III2. For Q-II1 and Q-II2, the details of hardware evaluation added in the appendix seem to check out. For Q-II3: In these figures, I hoped to figure out the computation burden added to the server as compared to FL-SSL schemes where the server is mainly aggregating the client models only. In MocoSFL since the server is first processing the latent features from the clients and then aggregating the models, the computation required for processing, backpropagation and aggregation could be substantial. I was hoping the authors would provide some numbers for the amount of processing happening on the server side as compared to FL-SSL. Overall, I raise my score to 6. Again, thanks for the hard work during rebuttal.

---

> > > ### Author Response · Authors · 2022-11-28
> > > **Sincere thanks for your review efforts and valuable feedback**
> > >
> > > Dear Reviewer bsEM,
> > >
> > > We want to express our appreciation for your comments and insights again. **Just a friendly reminder**: could you double-check whether the score of your review has been raised in the system? Because we still see "Score 5" in the current page.
> > >
> > > For your last question: following the same setting in Fig. 7, we quickly ran calculations on the server-side computation overhead for three schemes, which are listed in the table below. In MocoSFL schemes, the server needs to perform forward/backward computation and hence the computation is much higher than performing weight aggregation in FL. This is not a surprise and we think such a cost is practical given a powerful server. We will add a refined table to the final version of the paper.
> > >
> > > | **FLOPs comparison** | **Server**   | **Client**   |
> > > |----------------------|--------------|--------------|
> > > | **FL-SSL**           | **2.02e3 G** | **7.95e6 G** |
> > > | **MocoSFL**          | **1.35e7 G** | **2.19e3 G** |
> > > | **MocoSFL-TAResSFL** | **1.35e7 G** | **7.34e2 G** |
> > >
> > >
> > > Please kindly let us know if you have any concerns you find not fully addressed. We would be more than happy to include your suggestions in the final paper.
> > >
> > > Best,
> > >
> > > Authors

---

> ### Author Response · Authors · 2022-11-11
> **Response to Reviewer bsEM: (1/2)**
>
> **Q1:** Lack of Novelty: The novelty is limited as the work represents a combination of known methods.
>
> **Answer:** Thanks for the valuable comments. We address two important problems in SOTA FL-SSL schemes that make it impractical for cross-client applications. These are availability of large amount of data that is required for contrastive learning and the ability to process them in resource-constrained clients. The proposed MoCoSFL scheme addresses these problems by utilizing the key-storing mechanism in MoCo and Split Federated Learning (SFL) to support a smaller client-side model that reduces the client-side computations, latent vector concatenation that makes micro-batch training possible for clients and feature sharing that removes the requirement of a large amount of local data. Furthermore, we propose TAResSFL to address the communication overhead and privacy issues of MoCoSFL.
>
>   \
> **Q2:** Evidence to support the server can have a subset of validation data.
>
> **Answer:** Thank you for the suggestion. We provide additional evidence to support our assumption that the server can have a subset of validation data. For instance, in Lin et al. [1], the early stopping strategy in ensemble distillation requires the server to have some validation data, in Nagalapatti et al. [2], the irrelevant client mitigation method  heavily relies on the server to have validation data, in Wang et al. [3] FL’s non-IID performance is improved using available validation data and in Zhang et al. [4] the block-chain-based FL system relies on validation data for quality evaluation.
> However, we admit the assumption is not always practical and many existing works [5, 6] are opposed to such an assumption.We show that even if 0% validation data is available, our TAResSFL algorithm still achieves > 80% accuracy.
>
> **References:**
>
> [1] Lin et al. Ensemble Distillation for Robust Model Fusion in Federated Learning
>
> [2] Nagalapatti et al. Game of gradients: Mitigating irrelevant clients in federated learning
>
> [3] Wang et al. Optimizing Federated Learning on Non-IID Data with Reinforcement Learning
>
> [4] Zhang et al. Refiner: A reliable incentive-driven federated learning system powered by blockchain
>
> [5] Wu et al. Mitigating Backdoor Attacks in Federated Learning
>
> [6] Fung et al. Mitigating Sybils in Federated Learning Poisoning
>
>
>   \
> **Thank you again for your insightful questions. We reorganize other questions according to their category - I: Empirical Results, II - Hardware Related, III - Practicality**
>
> **I: Question on Empirical Results:**
>
> **Q-I1**: The obvious baselines FedEMA and FedU are missing from Table 2 and Table 3.
>
> **Answer:** In Table 2, FL-BYOL is a subset of FedEMA and it is the best among all techniques presented in FedEMA. FedU is inferior to FL-BYOL so we put that result in Appendix B.7 instead of in the main paper. We do not include them in Table 3 since their model achieves very low accuracy for more than  20 clients. Instead, we present FL-SSL cross-client accuracy results in Figure 1(b) as a motivation for the large data requirement. We show that they achieve very low KNN accuracy of just above 30% when the number of clients is 100.
>
>
>   \
> **Q-I2**: From Table 5, the results for Non-IID cases with Nc=5 or Nc =100 are better than IID cases which is counter-intuitive. Could the authors provide justification for this performance gain?
>
> **Answer:** We are also surprised to see that sometimes Non-IID cases have a little better performance than IID cases. We suspect there is a weak regularization effect when the data is non-IID. However, we do not have a clear way of proving it.
>
>   \
> **Q-I3**: Eq (2) finds that having a large batch-size can maintain hardness of negative samples, which seems to contradict the empirical results in Table 6 where increasing the batch size further to 400 degrades the performance. Reasons?
>
> **Answer:** The hardness is not the only reason affecting the accuracy - training dynamics can also be a factor. We found that using batch size 100, 200 can already fulfill the hardness requirement as indicated in Eq (2). When batch size is greater than 200, the total number of updates is less though the gradient is more “accurate”. Since we set the learning rate for all batch sizes to be the same - it justifies the small accuracy degradation.
>
>
>   \
> **Q-I4**: Did the authors implemented/reproduced the main results of baselines works e.g. FedEMA? It seems like the results are exactly as in the original paper.
>
> **Answer:** Yes, we generated results of FedEMA based on their open-sourced code for Figure 1. For Table 2, we took the original number from FedEMA paper after verifying its correctness (by inspecting and rerunning the provided code).
>
>
>
>
>
>   \
> (Continue in "Response to Reviewer bsEM: (2/2)")

---

> ### Author Response · Authors · 2022-11-11
> **Thanks to Reviewer bsEM**
>
> Thank you very much for the positive feedback and valuable comments. We hope the following clarifications can address your concerns. Please see "Response to Reviewer bsEM: (1/2)" and "Response to Reviewer bsEM: (2/2)" and kindly let us know whether we have adequately addressed your comments. We are looking forward to your feedback!

---

### Author Response · Authors · 2022-11-17
**Rebuttal Summary**

We sincerely thank all reviewers for their valuable comments and suggestions. We have made the following updates:

1. We remade Figure 5 by using arrows on x-axis and y-axis.
2. We have included details of overhead calculation in Table1 and Fig. 7 in Appendix A.5 section.
3. We included extensive results of MocoSFL on MobileNet and VGG architectures in Appendix B. 2.
4. We added extensive metrics (SSIM and PSNR) for Table 4 in Appendix B. 5.

We have revised our paper according to all the valuable comments and marked them in orange. Please kindly let us know if there is anything still not clear.

---

### Decision · Program_Chairs · 2023-01-20

**Decision:**

Accept: notable-top-5%

**Justification For Why Not Higher Score:**

N/A

**Justification For Why Not Lower Score:**

We have consensus this work is very solid and is the first to overcome several big limitations in previous systems

**Metareview: Summary, Strengths And Weaknesses:**

The paper introduces a federated self-supervised learning method, combining federated Split Federated Learning (SFL) and MoCo.
Reviewers were consistently positive about the paper, and liked several of the contributions made. This includes the ability to deal with large number of clients, each potentially having only a small amount of data (while heterogeneous), and limited compute capacity. The novel methodology is thoroughly evaluates, so could have practical impact.

Overall, reviewers all agreed that the paper is bringing novelty, is well-written and deserves acceptance.

We hope the authors will incorporate the several minor points mentioned by the reviewers during the discussions.


**Note From Pc:**

if the above contains the word "oral" or "spotlight" please see: "oral" presentation means -> notable-top-5% and "spotlight" means -> notable-top-25%. As stated in our emails, we are disassociating presentation type from AC recommendations